# The Impact of the Apelinergic System on the Cardiovascular System

**DOI:** 10.3390/ijms262010087

**Published:** 2025-10-16

**Authors:** Rafał Wyderka, Łukasz Osuch, Bogusława Ołpińska, Maria Łoboz-Rudnicka, Dorota Diakowska, Anna Leśków, Joanna Jaroch

**Affiliations:** 1Faculty of Medicine, Wroclaw University of Science and Technology, 50-370 Wroclaw, Poland; rafal.wyderka@pwr.edu.pl (R.W.); joanna.jaroch@pwr.edu.pl (J.J.); 2Department of Cardiology, Tadeusz Marciniak Lower Silesia Specialist Hospital, Emergency Medicine Center, Fieldorf 2, 54-049 Wroclaw, Poland; olpinskab@gmail.com (B.O.); marialoboz@o2.pl (M.Ł.-R.); 3Division of Medical Biology, Faculty of Nursing and Midwifery, Wroclaw Medical University, Chalubinskiego 4, 50-368 Wroclaw, Poland; dorota.diakowska@umw.edu.pl (D.D.); anna.leskow@umw.edu.pl (A.L.)

**Keywords:** apelinergic system, ELABELA, APJ receptor, cardiovascular diseases

## Abstract

The apelin–ELABELA–APJ axis, collectively known as the apelinergic system, has emerged as a key regulator of cardiovascular homeostasis. Acting through G-protein-coupled mechanisms, it modulates vascular tone, cardiac contractility, angiogenesis, fluid balance, and metabolism. Growing evidence indicates that dysregulation of apelinergic signaling contributes to the development and progression of atherosclerosis, hypertension, and heart failure. Experimental studies demonstrate cardioprotective actions of apelin and ELABELA, including anti-fibrotic, anti-inflammatory, vasodilatory, and pro-angiogenic effects, whereas some findings suggest context-dependent pro-atherogenic or vasoconstrictive roles. Clinical data show that circulating apelinergic peptides vary across cardiovascular conditions, being upregulated in acute coronary syndromes and diminished in chronic ischemic or hypertensive disease. In heart failure, early compensatory activation is followed by progressive depletion, and low ELABELA levels correlate with disease severity. Moreover, the apelinergic system may exert anti-arrhythmic effects through modulation of myocardial electrophysiology and structural remodeling. Novel synthetic APJ agonists and stabilized peptide analogs show promising preclinical efficacy in reducing cardiac remodeling, improving contractility, and lowering blood pressure. Altogether, the apelinergic pathway represents a multifaceted modulator and a promising therapeutic target in cardiovascular medicine, warranting further translational studies to elucidate its diagnostic and treatment potential.

## 1. Introduction

The apelin–ELA–APJ axis, known as the apelinergic system, has emerged as a multifunctional regulator of cardiovascular homeostasis. Acting through diverse molecular pathways, it modulates vascular tone, cardiac contractility, angiogenesis, fluid balance, and metabolic regulation [1]. Apelin, an endogenous ligand of the G protein-coupled APJ receptor, together with ELABELA, is widely expressed in the cardiovascular system, where it influences vascular function and cardiac performance. Although discovered relatively recently, the apelinergic system has attracted growing attention due to its diverse physiological and pathophysiological roles.

Accumulating evidence indicates that dysregulation of this axis contributes to the development and progression of cardiovascular diseases, including atherosclerosis, hypertension, and heart failure [2]. Preclinical and clinical studies suggest predominantly protective effects of apelinergic signaling—for instance, limiting infarct size, stimulating angiogenesis, counteracting maladaptive remodeling, and exerting anti-fibrotic and anti-arrhythmic actions. However, some reports describe opposite outcomes, such as pro-atherogenic activity in vascular smooth muscle cells, vasoconstrictive actions, or dual influence on cardiac hypertrophy. These discrepancies highlight the complex, context-dependent nature of apelinergic signaling and underscore the need for more precise characterization of its role in cardiovascular pathology.

Therapeutic perspectives include the development of stable apelin and ELABELA analogs and novel delivery systems. However, the short half-life of native peptides and incomplete understanding of long-term safety remain important challenges for their introduction into clinical practice

In this review, we summarize and discuss the current state of the art on the apelinergic system, integrating molecular, experimental, and clinical perspectives. Particular emphasis is placed on its contribution to ischemic heart disease, hypertension, hypertensive heart disease, heart failure, and arrhythmia. By synthesizing current evidence, this article aims to clarify the complex role of the apelinergic system in cardiovascular disease and to outline perspectives for clinical use.

## 2. Molecular Basis

The apelinergic system consists of the APJ receptor (a product of the *APLNR* gene) activated by two families of endogenous ligands: apelin and ELABELA (ELA) peptides [1].

The apelin family, which was discovered first, is a group of biologically active C-terminal fragments derived from post-translational processing of 77-amino acid prepropeptide, including apelin-36, apelin-19, apelin-17, apelin-13, and apelin-12 [1].

ELA, discovered more recently, plays a crucial role in embryonic development of the cardiovascular system [3], a fact which explains an apparent discrepancy between deficiency in the APJ receptor, causing severe cardiovascular malformations in APLNR-knockout mice, but not in apelin-knockout mice. However, as ELA is also expressed in adults, its role in cardiovascular system ranges beyond its embryonic development.

The overall physiological effects of apelin and ELA are similar; potencies and efficacies have been observed to vary between isoforms [1]. Regarding apelin, with biological activity of its isoforms being inversely proportional to their length, apelin-13 is considered the most potent, and due to its further modification to [Pyr1]-apelin-13, characterized by longer biological half-life, it is a predominant isoform in plasma [4].

The APJ receptor belongs to the family of G protein-coupled receptors. Despite sharing approximately 50% homology with the AT1 receptor, the APJ receptor is not activated by angiotensin [5]. The expression of APJ receptor is induced by stress, salt loading, water deprivation, and hypoxia [6].

The apelinergic system has a wide range of physiological and pathophysiological effects and is widely distributed throughout the body, including the cardiovascular system. Its distribution pattern in the cardiovascular system suggests a paracrine mode of signaling with ligand production localized mainly in the endothelium and receptor expression in cardiomyocytes, endocardial cells, and vascular smooth muscle cells (VSMCs) [7].

The complexity of apelinergic system results from a multiplicity of ligands binding the APJ receptor, diversity of G proteins interacting with the APJ receptor according to cell types, vast diversity of intracellular signaling pathways activated by the APJ receptor after binding its ligands, and furthermore, from frequent dependence of physiological outcomes of the APJ receptor activation on the prevailing physiological state [8]. As a result, the apelinergic system has a unique ability to respond to many chemical and physical stimuli, and thus to regulate many aspects of homeostasis, such as vascular tone, myocardial function, insulin sensitivity, inflammation, cell proliferation, migration and angiogenesis, body fluid homeostasis, neuroendocrine stress response, regulation of gastrointestinal, and immune functions [9,10,11,12,13,14,15].

APJ receptor interacts with G proteins, leading to the modulation of different intracellular signaling pathways upon ligand binding. As an example, through activation of the protein kinase B and protein kinase pathways, the apelinergic system is involved in the regulation of apoptosis, cell proliferation, neuroinflammation, and oxidative stress [16,17,18]. Enhancing nitric oxide (NO) production in the endothelium is an underlying mechanism of apelin-induced vasodilation [19]. ERK pathway activation results in cell proliferation, angiogenesis, and positive inotropic effect [1]. By modulating Ca transients and improving myofilament sensitivity to Ca^2+^, apelin enhances inotropy without increasing overall intracellular Ca^2+^ levels [19]. It has also been demonstrated that apelin signaling counteracts angiotensin II-mediated effects, suppressing the activation of pro-inflammatory pathways, inhibiting myocardial fibrosis and hypertrophy [20]. Figure 1 and Figure 2 illustrate the principal mechanisms through which apelin and ELABELA, acting via the APJ receptor, exert their cardiac and vascular effects, including the modulation of contractility, cardioprotection, vasodilation, and vascular tone regulation.

A growing body of research indicates that the apelin–ELA–APJ system plays an important role in regulating the cardiovascular system and atherosclerotic processes. The following sections present the current state of knowledge regarding the role of apelinergic system in selected cardiovascular conditions, as well as its potential therapeutic implications.

## 3. Current State of the Art

### 3.1. Ischemic Heart Disease

The role of apelinergic system in atherogenesis is unclear, with preclinical studies reporting both beneficial [21] and detrimental [22] effects. Apelin-13 downregulates the expression of lipoprotein lipase through the AOJ/PKCα/miR-361-5p signaling pathway in macrophage-derived foam cells, leading to inhibition of lipid accumulation and pro-inflammatory cytokine secretion and as a result, alleviating the atherosclerosis [23]. Another mechanism of regulating macrophage lipid metabolism comprises reducing lipid accumulation in foam cells by activating autophagy through the PI3K/Beclin-1 pathway [24]. While in macrophages, apelin exerts an anti-atherogenic and anti-inflammatory role; the actions of apelin on VSMC are mostly pro-atherogenic. Apelin-13 induces VSMC proliferation via Cyclin D 1 [25] and migration through matrix metalloproteinase (MMP)-2 activation [26], both considered critical processes in the pathogenesis of atherosclerosis. Additionally, in an animal ex vivo study, Cardoso dos Santos found that apelin promotes VSMC dedifferentiation from a contractile/differentiated towards a synthetic phenotype, a process characteristic of atherosclerosis development [27]. The discrepancy between those results might be due to a different impact of apelin on early atherosclerosis development versus late-established plaque development; the pleiotropic functions of the apelin/APJ system depend on cell type, tissue, and disease.

Myocardial infarction, in most cases resulting from the rupture of an atherosclerotic plaque, is one of the most serious clinical manifestations of atherosclerosis. Data on the apelinergic system in patients with myocardial infarction is scarce. Donmez et al. reported that patients in the acute phase of MI had ELA levels many times higher than healthy controls, and elevated ELA correlated with a poorer left ventricular ejection fraction [28]. Similarly, our group found a significant rise in apelinergic factors (ELA and AP-17) in patients with acute coronary syndrome (ACS) compared to controls and to those with chronic coronary syndrome (CCS) [29]. Du et al. also reported higher plasma ELA in patients with ACS versus controls without coronary artery disease [30]. This upregulation of apelinergic system could suggest a compensative mechanism in ACS.

However, in the study by Zhou et al. on 196 patients with ACS, plasma apelin concentrations sampled 2 h after ACS were significantly lower than in the control population [31]. The discrepancy may result from the moment of sample collection; therefore, further research is needed on the dynamics of concentrations of apelinergic system elements in ACS. Interestingly, in the study by Zhou et al., the apelin levels were significantly lower in the group with the ruptured plaque than in those with the nonruptured plaque; the plaque characteristics of the culprit lesion were assessed by an intravascular ultrasound imaging system [31], which would suggest an association of apelin/APJ axis with plaque vulnerability. Following myocardial infarction, an area of scar tissue forms in the myocardium, which initiates adaptation processes—molecular, cellular, and interstitial changes that manifest clinically as changes in size, mass, geometry, and function. Whether physiological adaptation to injury turns into maladaptation, which is called adverse cardiac remodeling, depends primarily on infarct size, but also on additional factors, including hemodynamic load conditions or neurohumoral activation [32]. The apelinergic axis may play a crucial role in the heart’s response to ischemia. Expression of apelin and the APJ receptor increases under hypoxic conditions associated with myocardial infarction [33]. APJ stimulation during acute infarction triggers cardiomyocyte survival pathways and promotes angiogenesis in the ischemic area, while suppressing pro-inflammatory cytokine release. Numerous experimental studies have confirmed the cardioprotective effects of both apelin and ELA in models of infarction and ischemia-reperfusion injury. In animal studies, administration of exogenous apelin (e.g., apelin-13) immediately after infarction-induced vasodilation via increased NO synthesis reduced infarct size and limited cardiomyocyte apoptosis [34]. In apelin-deficient mice, infarct size was larger, systolic function worse, and adverse post-infarct remodeling more pronounced compared to wild-type mice, whereas treatment with synthetic apelin analogs mitigated these deleterious changes [35]. Additionally, Jin et al. demonstrated that gene therapy increasing ELA expression enhanced angiogenesis in infarcted hearts, suggesting its potential use in regenerative strategies [36]. Interestingly, in our prospective study on 49 patients with STEMI anterior, patients presenting with an adverse cardiac remodeling had significantly higher apelin-13 and -17 levels one year post-MI [37]. In the multivariate analysis, only apelin-17 was independently associated with the occurrence of adverse cardiac remodeling, suggesting a strong compensatory apelinergic activation in patients with progressive remodeling. On the contrary, Krasniqi et al. reported a higher level of apelin-12 within seven days after MI correlated with low rates of MACE [38]. The apelinergic system exerts various pleiotropic cardioprotective effects: it inhibits excessive hypertrophy, reduces fibrosis (via anti-inflammatory and anti-TGFβ mechanisms), stimulates angiogenesis, and enhances metabolism and cellular survival [39]. In an animal study, treatment with apelin-13 alleviated Ang II-induced atherosclerosis through inhibition of Ang II intracellular signaling and promotion of NO production [21]. After MI, apelinergic signaling contributes to infarct size reduction: in rats, apelin-13 infusion for 5 days before ischemia/reperfusion induction by LAD occlusion (30 min) lowered infarct size from by 30%, in combination with losartan by 48% [40]. In rats, apelin-13 hindered mitochondrial permeability transition pore opening, which is responsible for cell death in ischemia/reperfusion injury [41]. In animal studies, Tatin et al. showed that after MI, the overexpression of apelin in the myocardium had a protective effect by preventing cardiac fibroblast activation, reducing the levels of TNFα and IL1β, and decreasing CD68+ macrophage infiltration in the ischemic heart [42]. In MI rats, apelin-13 administration for 4 weeks prevented cardiac fibrosis by inhibiting the increase in collagen I, collagen III, and TGF-β levels [43]. Intraperitoneal injection of apelin-13 for 4 weeks after LAD ligation-induced MI, reverted EMC-degrading enzymes MMP-2 and -9 in rat hearts, and reduced TGF-β and NF-kB pro-inflammatory mediators resulted in diminishing the histopathological damage and cardiac fibrosis percentage [44]. The apelinergic system performs as a functional antagonist to the angiotensin II–AT1 axis: apelin inhibits angiotensin II–stimulated pathways, limiting myocardial growth and extracellular matrix accumulation [8,45]. Thus, it is seen as a promising target to prevent adverse cardiac remodeling. In mice, chronic subcutaneous administration of metabolically resistant apelin-17 analog, LIT01-196, for 4 weeks post-MI reduced cardiac remodeling by limiting ventricular dilation, LV wall thinning, cardiac fibrosis, and by increasing vascular density and improving systolic performance of the left ventricle as assessed by echocardiography: increased left ventricular ejection fraction and fractional shortening [46]. Regarding the role of apelinergic syndrome in clinical presentation and complications of myocardial infarction, our recent study in patients with MI also showed that high ELA levels at the time of MI were associated with a higher incidence of atrioventricular conduction disturbances (AV blocks) over the following 12 months [46]. We also observed that elevated apelin-13 during acute MI correlated with a lower Q/QRS index after one year (Q/QRS index being a novel ECG indicator of post-infarct scar, developed by our team) [47]. These findings imply that apelinergic activation profiles during acute MI may influence arrhythmic complications along with myocardium healing processes.

Regarding the role of apelinergic syndrome in CCS, in a study performed by our group, levels of AP-17 and ELA were significantly lower in CCS than in healthy controls and ACS patients [29]. In a recent study on patients with triple-vessel coronary disease who underwent coronary artery bypass grafting, Rachwalik et al. showed that the serum levels of ELABELA and AP-17 were significantly lower, and the serum levels of APJ were significantly higher in the studied group than in healthy controls. Interestingly, a significant correlation was found between the serum and epicardial adipose tissue levels of ELABELA and APJ, while the tissue level of ELABELA correlated negatively with BMI, TCH, and LDL levels. This could suggest that a decrease in ELABELA production in epicardial adipose tissue might contribute to an impaired lipid metabolism [48]. Aksakal et al. examined patients with chronic obstructive pulmonary disease (COPD) and showed that patients with COPD and CAD presented with lower serum apelin levels compared to those without CAD. ROC analysis indicated high sensitivity and specificity for apelin-13 and -36 in predicting CAD in COPD patients [49]. This may be of particular clinical significance as symptoms of CAD and COPD often overlap, and CAD is often underdiagnosed in patients with COPD. Regarding coronary chronic total occlusions, Yavuz et al. found significantly lower serum ELA levels in patients with stable angina and CTO compared to those with patent coronaries [50]. Low ELA levels also correlated with poor collateral circulation—patients with Rentrop grades 0–1 had significantly lower ELA than those with well-developed collaterals (Rentrop 2–3). In this study, a multivariate analysis of low ELA alongside elevated NT-proBNP was an independent predictor of CTO [50]. These pilot findings suggest ELA deficiency may be linked to more advanced coronary disease and reduced collateral formation capacity in chronic myocardial ischemia. These differences suggest that in CCS, apelinergic axis activity is diminished, likely due to progressive endothelial damage and risk factor-induced inhibition of cardioprotective peptide expression. Therefore, lower circulating apelin and ELA in stable CAD may indicate loss of vascular protection, promoting disease progression. This aligns with earlier observations of lower apelin in stable CAD patients versus healthy individuals [29]. Chronic vascular exposure to damage (e.g., prolonged hypertension) may reduce ELA and apelin synthesis/secretion by dysfunctional endothelium, weakening apelinergic protection in chronic CAD. Table 1 summarizes clinical and experimental studies demonstrating the diverse mechanisms by which the apelinergic system influences the pathogenesis and progression of ischemic heart disease.

### 3.2. Hypertension and Hypertensive Heart Disease

Long-term hypertension causes organ damage and ultimately leads to the development of cardiovascular, cerebrovascular, and clinical kidney disease. Hypertension-induced cardiac remodeling characterized by ventricular hypertrophy, fibrosis, and impaired angiogenesis eventually leads to myocardial dysfunction and heart failure. It is a complex process in which the apelinergic system plays an important role [8].

Several preclinical studies provide data on dynamic alterations in apelinergic system components in hypertensive rats, showing compensatory upregulation of the APJ receptor and reduced levels of apelin in spontaneously hypertensive rats [51,52]. In rats with an acquired, acute hypertension produced by placing a clip around renal artery, the investigators observed that APJ receptor expression in the heart was significantly reduced in the acute phase of hypertension and partially recovered in the chronic phase, while apelin level reduced in both phases, with a significant decline only in chronic phase [53]. Similarly, aortic APJ receptor expression was reduced in both phases, with a significant decline only in chronic phase. APJ receptor downregulation in the acute phase of renal hypertension may be associated with high levels of angiotensin II, with its inhibitory effect reduced in the chronic phase.

In human studies, hypertensive patients have lower apelin levels than healthy controls [54]. Importantly, Ma et al., in an elegant prospective study, showed that a deficiency in the apelinergic signaling may be one of the factors contributing to the progression of hypertensive heart disease. Among patients with hypertension, those with lower plasma ELABELA levels presented lower left ventricle ejection fraction, higher biomarkers of myocardial damage like BNP and troponin, and higher rate of heart failure hospitalizations [55]. In a recent study, Tian et al. showed lower serum levels of ELABELA in patients with hypertension, and lower levels of ELABELA in patients with malignant hypertension as compared to benign hypertension [56]. In untreated hypertensive patients, Ye et al. showed an independent association of low apelin level with left ventricular hypertrophy [57].

The role of the apelinergic system in the pathogenesis of hypertension and hypertensive heart disease is complex and not entirely understood. In terms of beneficial effects, apelinergic system causes vasodilatation—administration of apelin-36 and Pyr1-apelin-13 to humans caused NO-dependent arterial vasodilation [58]. Results of Wang et al.’s study suggest that ELABELA causes vasorelaxation in a less endothelium-dependent manner—in mice, pretreatment of L-NAME (NO production inhibitor) did not abolish the relaxation effect of ELABELA, indicating that NO is not required for the effect [59]. In the rodent model, Chen et al. demonstrated that ELABELA overexpression prevented DOCA/salt-induced hypertension by blocking NADPH oxidase/ROS/NLRP3 signaling pathway in the kidney, which suggests a potential role of ELABELA in treating salt-sensitive hypertension [60].

However, there are reports in the literature on apelinergic syndrome contributing to the development of hypertension. Zhang et al. reported increased apelin expression in the rostral ventrolateral medulla in hypertensive rats, suggesting that the apelinergic system contributes to sympathetic overdrive and subsequently to hypertension [61]. Xu et al., in a study on hypertensive rats, showed that by stimulating VSMC proliferation, apelin-13 also contributes to vascular remodeling in hypertension. It would suggest that the downregulation of apelin in hypertensive rats might be a compensatory protective mechanism to avoid excessive VSMC proliferation and subsequent maladaptive vascular remodeling [52]. Animal studies also indicate a vasoconstrictor action of apelins: chronic infusion of apelin-13 into a paraventricular nucleus increases levels of plasma norepinephrine and arginine-vasopressin, and subsequently induces hypertension in normotensive rats [61,62]. The apelin/APJ system has opposite effects on vascular tone when acting on endothelial cells and VSMC. On one hand, apelin promotes vasodilation through NO production when binding to endothelial APJ receptors. On the other hand, apelin binding to VSMC promotes vasoconstriction through inhibition of BKCa channel and subsequent smooth muscle contraction [62,63].

It is also important to underscore a complex interplay between the apelinergic system and RAAS system, with their structural homology and overlapping tissue distribution but contrasting functional actions. In experimental studies, apelin infusion-blunted Ang-II-induced vasoconstriction [64], inhibited Ang II activation of pro-fibrotic factors like plasminogen activator inhibitor-1 and activation of pro-fibrotic signaling pathways in VSMC and cardiac fibroblasts. Administration of pyr1-apelin-13 prevented the progression of Ang II-mediated hypertension, cardiac hypertrophy, fibrosis, and dysfunction in mice with elevated ACE2 levels [65]. Angiotensin-converting enzyme 2, which plays a protective role in the cardiovascular system by degrading Ang II, is reported to cleave certain apelin isoforms [1], while the apelin/APJ signaling upregulates ACE2 expression [66]. This indicates that both systems regulate each other. Due to many internal interactions between the apelinergic system and RAAS system, a concept of renin–angiotensin–aldosterone–apelinergic system has been recently introduced [67].

The apelinergic signaling effect on the development of cardiac hypertrophy, a key component of hypertension-induced cardiac remodeling, is bifunctional [68]. On one hand, the apelinergic system alleviates Ang II-induced hypertrophy through inhibiting the expression of TGF-beta [57] and oxidative stress-induced hypertrophy through inhibiting the generation of H202 [69]. On the other hand, in a rodent study, Li et al. showed that apelin-13 promotes hypertrophy via upregulating the levels of ROS in VSMC and promoting oxidative stress when administered in non-physiological dose [70]. Interestingly, stretch-mediated β-arrestin activation of APJ receptor leads to pro-hypertrophic response; however, apelin treatment was shown to prevent stretch-induced cardiac hypertrophy via a G-protein-dependent mechanism [71]. Animal studies indicate cardioprotective effects of APJ signaling in pressure-overload models. Continuous infusion of ELABELA in mice subjected to pressure overload by transverse aortic constriction (TAC) reduced mRNA expression of genes associated with cardiac hypertrophy, such as BNP (brain natriuretic peptide) and β-Myhc (β-myosin heavy chain), and histological analysis revealed that ELA treatment reduced the area of cardiac fibrosis. Consistently, ELABELA treatment downregulated the expression of pro-fibrotic genes: TGFβ2, Latent TGFβ binding protein 2 (Ltbp2), Periostin, and Collagen 8a. In TAC mice, ELABELA also downregulated the expression of angiotensin-converting enzyme 2 [20].

To conclude, the apelinergic system plays an important, yet not fully elucidated role in the pathogenesis of hypertension and hypertensive heart disease. Further studies are needed to precisely define its actions on the cardiovascular system and to explore its therapeutic potential in hypertensive heart disease. Table 2 summarizes experimental and clinical evidence on the apelinergic system in hypertension, highlighting its roles in hemodynamic regulation, vascular remodeling, interaction with the renin–angiotensin–vasopressin axis, renal effects, and its clinical associations with hypertensive organ damage.

### 3.3. Heart Failure

The apelinergic system is involved in the development of heart failure, as it influences the pathophysiology of two main diseases underlying heart failure, i.e., hypertension and coronary artery disease; but importantly, the apelinergic system also influences the pathophysiology of heart failure as a pump—not only being a strong positive inotrope but also significantly influencing the body’s water balance, and therefore fluid retention.

The apelinergic system is involved in the pathogenetic processes of heart failure, with plasma apelin levels increasing in the early stages of heart failure but decreasing in more advanced ones [7,77]. After cardiac resynchronization therapy, an improvement in New York Heart Association (NYHA) class and left ventricular ejection fraction is paralleled by restoration of normal plasma apelin concentration [78]. In human cardiac tissues, APJ receptor density was significantly decreased in the left ventricle of patients with dilated cardiomyopathy, but apelin levels remained unchanged, suggesting that lowered receptor density may limit the positive inotropic influence of apelin [79]. The involvement of the apelinergic system may vary depending on the etiology of heart failure. In patients with advanced heart failure due to idiopathic dilated cardiomyopathy, APJ receptor mRNA levels within the left ventricle were reduced by 44%, but were unaltered in patients with ischemic cardiomyopathy [79,80].

Studies indicate also a diagnostic potential of APJ receptor agonists in heart failure patients. A recent study by Chunju Liu et al. demonstrated significantly lower plasma ELABELA levels but higher plasma levels of apelin in HF patients than in non-HF patients [81]. Moreover, ELA levels decreased progressively with increasing heart failure severity (higher NYHA class, lower ejection fraction) and low ELA levels negatively correlated with cardiac function parameters (higher BNP, larger left ventricular size and volume, thicker posterior wall, lower ejection fraction), suggesting that ELA deficiency is associated with more severe cardiac dysfunction. On the contrary, plasma apelin was not affected by the severity stratified by NYHA grade, the reduction in LVEF, or the sustained increase in plasma BNP in HF patients. Notably, in the study by Liu et al., the diagnostic ability of ELA for identifying heart failure was high—the area under the ROC curve for heart failure diagnosis was 0.835 (at 95% specificity), outperforming both apelin (AUC 0.673) and ejection fraction alone (0.612) [81]. These findings indicate that ELA may serve as a novel biochemical marker of heart failure, complementing classical markers in diagnosis and risk stratification.

Apelin is one of the most potent positive inotropic endogenous peptides [82]. In a study by Kuba et al., aging apelin-knockout mice displayed progressive impairment of cardiac contractility in the absence of histological abnormalities. Furthermore, in a pressure-overload-induced heart failure model, apelin-knockout mice also developed severe impairment in heart contractility [83]. The inotropic action of apelin is independent of angiotensin II, endothelin-1, catecholamines, or nitric oxide release [84]. In heart failure patients, intravenous administration of apelin increased cardiac output and left ventricle ejection fraction while reducing vascular resistance [85]. In rats, ELABELA increased left ventricular contractility, while simultaneously reducing arterial pressure [86]. Several pathophysiological pathways were postulated to be engaged in inotropic action of apelinergic system, like the PLC/PKC pathway or ion transport modulation—apelin signaling modulates Ca^2+^ transients and enhances myofilaments Ca^2+^ sensitivity, contributing to contractility without deleterious consequences of increased overall intracellular Ca^2+^ levels [87].

Apelin/APJ receptor axis promotes vasodilatation through NO release from endothelial cells, subsequently reducing vascular resistance, lowering pre- and afterload of the left ventricle, and as a consequence, improving its filling pattern. Thus, apelinergic system uniquely combines positive inotropic and vasodilatory actions, leading to cardiac unloading through the Frank–Starling mechanism (higher output with reduced preload and afterload) [2]. The cardioprotective actions of apelin/APJ receptor axis also result from the regulation of RAS system and inhibit the detrimental influence of angiotensin II [65].

Apelin also shows a positive chronotropic action, increasing conduction velocity in cardiomyocytes and inducing a shortening of action potential in atrial myocytes [72,88].

Furthermore, the apelin–ELA–APJ axis is involved in body fluid homeostasis, enhancing diuresis via a central and direct renal mechanism [89]. The central mechanism involves the inhibition of arginine-vasopressin release [75], while the renal mechanism involves counteracting the antidiuretic effect of arginine-vasopressin mediated by type-2 vasopressin receptors [76]. In animal studies, apelin-13 prevented the expression of aquaporin -2 in the kidney and subsequently promoted excreting water without electrolyte loss [73]. Hus-Citharel et al. proved another mechanism of diuretic effect of apelin-17 through vasorelaxation of glomerular afferent arterioles [74].

Japp et al. established that acute apelin administration in patients with heart failure causes peripheral and coronary vasodilatation and increases cardiac output. Intracoronary bolus of apelin-36 increased coronary blood flow and reduced peak and end-diastolic left ventricular pressures while systemic infusions of (Pyr1)-apelin-13 increased cardiac index and lowered mean arterial pressure and peripheral vascular resistance [85]. Barnes et al. showed sustained cardiovascular effects (increase in cardiac index) of prolonged infusions of (Pyr1)-apelin-13 in heart failure patients [90]. Importantly, the apelinergic system has been reported to have also beneficial effects in doxorubicin-induced cardiotoxicity [91]. The study by Buczma et al. performed on doxorubicin-treated rats showed that lower doses of apelin and ELABELA prevented the prolongation of QT and QTc intervals induced by doxorubicin, and LV systolic function parameters improved in the DOX-treated groups that were simultaneously administered to APJ agonists [92]. The authors concluded that apelin and ELABELA could be regarded as potential cardioprotective agents against anthracycline-induced cardiotoxicity.

In summary, activation of the apelinergic system exerts multifaceted protective effects in heart failure—increasing systolic capacity and cardiac output, lowering arterial pressure, reducing cardiac load, and suppressing adverse remodeling processes (fibrosis, hypertrophy), as well as facilitating fluid clearance.

With the growing understanding of the role of the apelin–ELA–APJ system in heart failure pathophysiology, it is increasingly seen as a potential therapeutic and diagnostic target. Clinical studies are already underway to explore the therapeutic use of ELA and apelin in both acute and chronic heart failure—for instance, ELA is being proposed as a preventive early biomarker and a novel cardiotropic agent for heart failure treatment [93]. While further research is necessary, current findings offer hope that apelinergic modulation may become a future treatment strategy for heart failure. Table 3 presents clinical and experimental studies showing how the apelinergic system modulates heart failure through effects on cardiac contractility, hemodynamics, fibrosis, and hypertrophic signaling.

### 3.4. Arrhythmias

There is growing evidence that the apelinergic system may exert beneficial effects on cardiac rhythm disorders. Although the exact anti-arrhythmic mechanisms are not yet fully understood, studies suggest direct electrophysiological effects of apelin on the heart. In the context of atrial fibrillation (AF), the most common supraventricular arrhythmia, the apelinergic system plays an important role in electrical and structural remodeling of the atria. Early studies showed that apelin administration accelerates conduction in cardiac muscle: Farkasfalvi et al. observed in cardiomyocyte cultures that apelin induces phosphorylation of the sarcolemmal Na^+^/H^+^ exchanger (NHE), which increases conduction velocity and prevents slow depolarization waves that favor arrhythmogenesis [88]. This directly supports apelin’s anti-arrhythmic role via improved conduction (faster conduction may prevent re-entry phenomena). Other studies indicate that apelin prolongs atrial refractory period and counteracts angiotensin II–induced pro-arrhythmic effects, thus preventing AF initiation in experimental models [96]. Moreover, apelin strongly inhibits angiotensin II–induced atrial fibrosis by suppressing activation of the TGF-β/Smad pathway, thereby reducing the electrical and structural substrate for arrhythmogenesis [97].

Clinical studies have shown that both plasma apelin levels and local apelin expression in atrial tissue are reduced in patients with AF. Similarly, lower ELA levels have been reported in hypertensive patients with AF, compared to hypertensive patients without AF [98]. Interestingly, low apelin levels are associated not only with AF occurrence but also with its recurrence. Falcone et al. demonstrated that reduced apelin concentrations had surprisingly high sensitivity as a risk marker for AF in patients post-cardioversion or ablation [99]. There is also emerging evidence suggesting that boosting apelinergic activity may be beneficial for arrhythmia. The authors suggest that raising circulating apelin levels in AF patients may represent a future therapeutic strategy, and apelin may serve as a useful biomarker for AF detection and prognosis [99].

Apelinergic system components, ELA and APJ, may act anti-arrhythmically by affecting electrophysiology (accelerating conduction, prolonging refractoriness) and myocardial structure (inhibiting fibrosis and hypertrophy). In mice subjected to chronic intense exercise, apelin administration prolonged atrial refractory periods and reduced the incidence and duration of induced atrial arrythmias [96].

Beyond AF, the system’s role in other arrhythmias and conduction disorders is investigated. For example, as previously mentioned, high ELA levels during acute MI have been associated with later AV conduction disturbances, suggesting a link between apelinergic axis activation and the cardiac conduction system [47]. Conversely, apelin may help protect against dangerous ventricular arrhythmias by improving perfusion and left ventricular function, ensuring cardioprotective effect which reduces susceptibility to post-infarction ventricular arrhythmias. Apelin administered in MI mice shortened the action potential duration by increasing IK1 current-density via PI3-kinase-dependent signaling pathways and shortened QTc and QT interval induced by MI [100]. Overall, the apelinergic system appears to be a promising target for the prevention and treatment of both supraventricular and ventricular arrhythmia, although more clinical research is needed. Table 4 outlines clinical and experimental findings, indicating that the apelinergic system modulates electrophysiology and conduction, linking altered apelin or ELABELA levels with atrial fibrillation, conduction disorders, and arrhythmia recurrence, thus suggesting anti-arrhythmic potential.

### 3.5. Therapeutic Perspectives

The broad spectrum of cardioprotective effects associated with the apelin–ELA–APJ system makes it an attractive target in the search for novel therapies for cardiovascular diseases [101]. In conditions such as heart failure, myocardial infarction, and hypertension, enhancing apelinergic axis activity appears to be beneficial. However, there are significant challenges: the natural APJ ligands are peptides with a very short plasma half-life (on the order of minutes), which limits their direct pharmacological application—they require continuous intravenous infusion [94]. Therefore, considerable efforts have been directed toward developing modified analogs of apelin and ELA with improved stability and bioavailability. In recent years, several new peptide-based APJ agonists have been developed. For example, adding an N-methyl-arginine residue to the apelin-17 sequence extended its plasma half-life while retaining its hypotensive effects [40].

In animal studies, two promising apelin-17 analogs, designated P92 and LIT01-196, demonstrated exceptionally long plasma half-lives (thanks to chemical modifications, including the addition of a fluorocarbon chain in LIT01-196) and very high affinity for the APJ receptor [40]. Importantly, both analogs acted as full receptor agonists, activating all major APJ signaling pathways, and showed beneficial effects in vivo. They were shown to inhibit vasopressin release, increase diuresis, and significantly reduce blood pressure [40]. Notably, LIT01-196 exhibited strong and long-lasting antihypertensive effects in hypertensive rats and has been proposed as a potential long-acting next-generation antihypertensive agent [40]. Recently, another method of administration was tested—apelin 13, encapsulated in microparticle-embedded patch, was delivered epicardially in mice and protected from MI-induced cardiac dysfunction and adverse remodeling [95].

## 4. Limitations and Future Research Directions

Despite these exciting advances, the clinical implementation of apelinergic system modulation, especially targeted therapies, faces multiple challenges and uncertainties. The main limitation lies in the pharmacokinetics of natural ligands, which have short half-lives and require parenteral administration. Efforts to develop oral formulations (small-molecule agonists) are promising but still require further optimization for efficacy and safety in humans. A detailed assessment of the adverse effect profile of APJ stimulation is also essential. Apelin lowers blood pressure, which may limit dosing in certain situations (e.g., patients with hypotension due to advanced heart failure) [16]. On the other hand, this effect is desirable in hypertension or during the early phase of acute heart failure with elevated blood pressure. Preliminary animal studies suggest that APJ activation does not induce the arrhythmias or accelerate remodeling typically associated with conventional inotropes; on the contrary, it may exert protective effects. Nonetheless, the long-term consequences of chronic apelinergic stimulation in humans remain unknown.

## 5. Conclusions

The apelin–ELA–APJ system is emerging from current research as a key, multifunctional modulator of the cardiovascular system. In both acute (such as myocardial infarction, atrioventricular blocks, or arrhythmias) and chronic conditions (chronic heart failure, cardiac remodeling, hypertension), apelinergic signaling exerts protective effects through various mechanisms: reducing ischemic injury, improving perfusion and cardiac contractility, inhibiting pathological remodeling (including hypertrophy and fibrosis), exerting anti-arrhythmic effects, and beneficially modulating the cardiac conduction system. Preclinical studies consistently indicate that stimulation of the APJ receptor confers cardioprotective benefits—reducing infarct size, improving post-ischemic cardiac function, unloading the failing heart, and preventing further damage. The natural agonists of this receptor, apelin and ELA, thus represent some of the most promising endogenous cardioprotective molecules discovered in recent decades. Translating this knowledge into clinical practice is the next challenge. Initial therapeutic trials (apelin infusions in patients) have shown hemodynamic improvements without serious side effects, but longer-acting drug formulations are needed for sustained effects. Advances in molecular biology have already led to the development of the first APJ analogs and small-molecule agonists with improved pharmacological properties, which are currently undergoing preclinical and clinical evaluation. If proven effective and safe, these agents might revolutionize the treatment of heart failure, acute cardiac injury, and resistant hypertension. Furthermore, profiling of apelinergic biomarkers (apelin and ELA levels) may aid in prognostication of cardiovascular diseases—e.g., by identifying high-risk patients in need of more intensive therapy.

In summary, modulation of the apelin–ELA–APJ system represents a promising avenue for both research and therapy in modern cardiology. Integrating new scientific advances in this field may, in the future, lead to novel diagnostic and treatment strategies, offering tangible benefits for patients with heart disease.

## Figures and Tables

**Figure 1 ijms-26-10087-f001:**
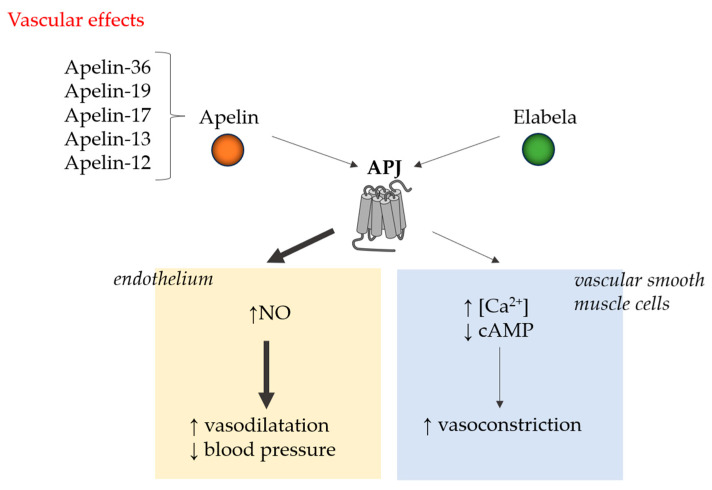
The vascular effects of apelin or elabela depend on the condition of the vascular bed. When endothelium is intact, the NO-mediated action outweighs the effect of increased Ca^2+^, resulting in vasodilation (thicker arrows in the diagram). ↑: increase; ↓: decrease.

**Figure 2 ijms-26-10087-f002:**
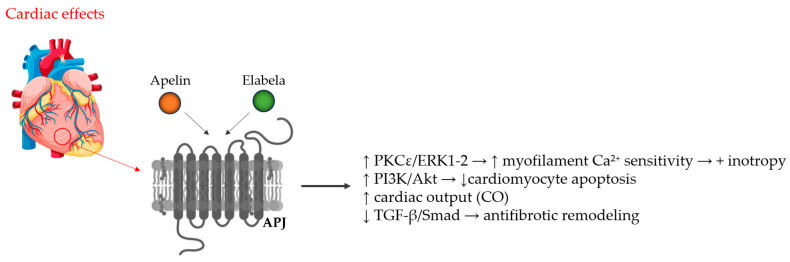
The cardiac effects of apelin or elabela after binding to the APJ (APLNR) receptor on the cardiomyocyte membrane. APJ signaling engages PKCε (protein kinase C isoform ε) and ERK1/2 (extracellular signal-regulated kinase), increasing myofilament Ca^2+^ sensitivity and thereby producing a positive inotropic effect; PI3K/Akt (phosphatidylinositol 3-kinase/protein kinase B) activation supports cardiomyocyte survival; inhibition of the TGF-β/Smad (Transforming Growth Factor-beta/Sma- and Mad-related protein) pathway contributes to anti-fibrotic remodeling.

**Table 1 ijms-26-10087-t001:** The role of apelinergic system factors in ischemic heart disease.

Mechanism of Action and Examined Cells/Models	Result of Action	Reference
Post-MI patients; apelinergic components vs. LV remodeling.	Apelinergic system associates with adverse LV remodeling.	Wyderka et al. [47]
Rat MI model; [Pyr1]-apelin-13 post-infarct.	Improved function via ↑ neovascularization and angiogenic factors.	Azizi et al. [33]
Myocardial I/R models; ELABELA→PI3K/AKT.	↓ apoptosis, fibrosis; mitigated mitochondrial dysfunction.	Yu et al. [35]
STEMI patients; plasma ELABELA on day 1.	↑ ELABELA; correlation with LV function biomarkers.	Dönmez et al. [28]
MI models; ELABELA gene therapy.	Promoted angiogenesis after MI.	Jin et al. [36]
Patients with CCS/ACS; plasma apelin/ELA.	Altered circulating apelinergic peptides in CAD/ACS.	Diakowska et al. [29]
ACS patients; ELABELA vs. coronary severity.	Lower ELABELA associated with greater angiographic severity.	Du et al. [30]
THP-1 foam cells; APJ/PKCα/miR-361-5p.	↓ LPL expression → anti-foam cell (anti-atherogenic).	Zhang et al. [23]
Foam cell formation; Class III PI3K/Beclin-1-mediated autophagy.	Apelin-13 activates autophagy → impedes foam cell formation.	Yao et al. [24]
Intimal SMCs; apelin expression and phenotypic transition.	Promotes SMC phenotypic switching (atherosclerosis).	Cardoso et al. [27]
ACS; apelin as marker of stenosis/plaque stability.	Apelin associated with coronary stenosis and plaque stability.	Zhou et al. [31]
Multivessel CAD; serum and epicardial adipose apelin/ELA.	Profiles of apelin/ELA in CAD and epicardial fat.	Rachwalik et al. [48]
STEMI; apelin-12 effects on troponin and MACE.	Apelin-12 influences biomarker profile and MACE risk.	Krasniqi et al. [38]
Post-MI; lymphatic endothelium remodeling under apelin.	Apelin modulates pathological lymphatic remodeling after MI.	Tatin et al. [42]
Rodent I/R; apelin-13→RISK–GSK-3β–mPTP.	Cardioprotection with smaller infarct.	Yang et al. [41]
Mouse atherosclerosis and aneurysm models; apelin vs. Ang II.	Apelin antagonizes Ang II; reduces atherosclerosis/aneurysm.	Chun et al. [21]
Atherogenesis under oxidative stress; requirement for apelin/APJ system.	Deficiency worsens lesions.	Hashimoto et al. [22]
COPD patients assessed for CAD using apelin.	Explores diagnostic value of apelin in CAD among COPD.	Aksakal et al. [49]

**Table 2 ijms-26-10087-t002:** The role of apelinergic system factors in hypertension.

Mechanism of Action and Examined Cells/Models	Result of Action	Reference
Adult CV system; ELA endogenous APJ agonist; PAH models; exogenous ELA administration.	Compensates for downregulated expression; improves hemodynamics in PAH.	Yang et al. [41]
Conscious rats; venous dilator effect of apelin.	Venodilation consistent with afterload/preload reduction.	Cheng et al. [72]
CNS apelinergic signaling in rat brain.	Central regulation of BP/HR (pressor/tachycardic actions).	Reaux et al. [10]
SHR; APJ upregulation → VSMC proliferation via autophagy.	Contributes to vascular remodeling in HTN.	Xu et al. [52]
2K1C Goldblatt hypertensive rats; myocardium/aorta APJ expression.	Hypertension alters APJ expression in heart and aorta.	Najafipour et al. [53]
C57BL/6J mice; Ang II-induced HTN/fibrosis; apelin administration.	Apelin protects vs. Ang II-induced HTN and cardiovascular fibrosis.	Siddiquee et al. [64]
Mouse models; apelin opposes Ang II-mediated remodeling/dysfunction.	Apelin negatively regulates Ang II effects.	Zhang et al. [65]
Hypertensive patients; apelin levels across anti-HTN drugs.	Differences in apelin with therapy; clinical association.	Hemmati et al. [54]
Concept/mechanistic study linking APLNR–VEGF–nNOS in HHD.	Suggests vascular footprint underlying hypertensive heart disease.	Iliev et al. [51]
PVN in SHR; apelin-13/APJ → sympathetic activation and vasopressin release.	Contributes to elevated BP via central mechanisms.	Zhang et al. [65]
Cerebral artery SMC; apelin-13 inhibits BK channels via PI3K.	Affects vascular tone; potential cerebrovascular constriction.	Modgil et al. [63]
Untreated HTN patients; serum apelin vs. LVH.	Lower apelin associated with LV hypertrophy.	Ye et al. [57]
Cardiac hypertrophy models; apelin activates catalase/antioxidants.	Prevents oxidative stress-linked cardiac hypertrophy.	Foussal et al. [69]
Kidney collecting duct cells; AQP2 trafficking vs. vasopressin.	Apelin antagonizes vasopressin → aquaresis.	Boulkeroua et al. [73]
Rat kidney; glomerular hemodynamic effects of apelin.	Renal vasorelaxation affecting diuresis/BP.	Hus-Citharel et al. [74]
Hypothalamus; apelin inhibits vasopressin neuron activity/release.	Potent diuretic effect counteracting vasopressin.	De Mota et al. [75]
Rat collecting duct; apelin–vasopressin receptor cross-talk.	Reduces vasopressin-induced water reabsorption.	Hus-Citharel et al. [76]
VSMC proliferation via Jagged-1/Notch3 after apelin-13.	Promotes VSMC proliferation (remodeling).	Li et al. [70]
VSMC migration via PI3K/Akt/FoxO3a/MMP-2.	Enhances VSMC migration (remodeling).	Wang et al. [59]
Hypertensive patients; serum ELA vs. renal damage progression.	Decreased ELA linked to hypertensive renal injury.	Tian et al. [56]

**Table 3 ijms-26-10087-t003:** The role of apelinergic system factors in heart failure.

Mechanism of Action and Examined Cells/Models	Result of Action	Reference
Cohort/observational; plasma ELABELA as HF screening indicator.	Lower ELABELA linked with HF; screening potential.	Liu et al. [93]
Human study; apelin infusion in healthy and chronic HF.	↑ CO; vasodilation; potential HF benefit.	Japp et al. [58]
Small-molecule APJ agonism; CV response in preclinical/early settings.	Hemodynamic effects consistent with APJ activation (↑ inotropy/vasodilation).	Ason et al. [94]
Hypertensive HF patients; plasma ELA predicts MACE.	Lower ELA predicts adverse events.	Ma et al. [55]
In vivo models (normal and failing hearts); apelin inotropy.	Positive inotropic effects demonstrated.	Berry et al. [82]
Apelin gene-deficient mice; aging/pressure overload.	Apelin deficiency → impaired contractility with stress.	Kuba et al. [83]
Experimental models; apelin regulates cardiac contractility.	Positive inotropy/contractile modulation.	Szokodi et al. [84]
Murine hypertrophy models; APJ dual receptor signaling.	APJ mediates protective vs. maladaptive hypertrophic signaling.	Scimia et al. [71]
Mouse; tissue-specific apelin/APJ control in hypertrophy→HF transition.	Controls hypertrophy and contractility.	Parikh et al. [87]
Human HF and atherosclerosis; modulation of apelin/APJ.	Altered apelin/APJ in HF and IHD.	Pitkin et al. [79]
MI-induced HF; apelin-13 inhibits PI3K/Akt-linked oxidative stress.	Alleviates cardiac fibrosis.	Zhong et al. [43]
Post-MI remodeling; apelin-13 anti-fibrotic effects.	Protects against MI-induced myocardial fibrosis.	Zhang et al. [44]
Apelin-17 analog LIT01-196 post-MI HF models.	Reduces dysfunction and remodeling after MI.	Girault-Sotias et al. [46]
Human cardiac dysfunction; apelin as endogenous inotrope.	Demonstrates inotropic role in human HF context.	Chen et al. [7]
HF patients; circulating and cardiac apelin levels.	Altered apelin levels associated with HF.	Foldes et al. [77]
CRT in HF; plasma apelin changes post-therapy.	CRT increases plasma apelin.	Francia et al. [78]
Post-MI mice; microparticle-mediated apelin delivery.	Improves heart function post-MI.	Tang et al. [95]

**Table 4 ijms-26-10087-t004:** The role of apelinergic system factors in arrhythmias.

Mechanism of Action and Examined Cells/Models	Result of Action	Reference
Post-MI patients; conduction disorders vs. apelinergic profile.	Apelinergic system influences conduction disorders.	Wyderka et al. [47]
Neonatal rat cardiomyocytes; electrophysiology and contractility with apelin.	Modulated electrophysiology supporting anti-arrhythmic potential.	Farkasfalvi et al. [88]
Hypertensive patients with AF; plasma ELABELA.	Lower ELABELA associated with AF in HTN.	Ma et al. [98]
Persistent AF patients; apelin plasma vs. recurrence.	Lower apelin predicts AF recurrence after cardioversion.	Falcone et al. [99]
Atrial myocytes; apelin regulation of electrophysiology.	Modulates atrial electrophysiological properties.	Cheng et al. [72]

## Data Availability

No new data were created or analyzed in this study. Data sharing is not applicable to this article.

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
