# Peer review of "The Impact of the Apelinergic System on the Cardiovascular System"

_ijms, 2025, doi:10.3390/ijms262010087_

Round 1

Reviewer 1 Report

Comments and Suggestions for Authors

This study describes the role of the apelinergic system in the cardiovascular system. However, a recent review article has addressed the relevant topics in greater detail and depth (Cardiovascular Research 2023;119:2683–2696), thereby diminishing the novelty and significance of this manuscript.

Author Response

We sincerely thank you for your valuable review. Indeed, the article published in Cardiovascular. Research is a good source of knowledge about the apelinergic system, and therefore we have added information about it to our manuscript. These article presents the state of knowledge on the apelinergic system of 2023; in our opinion, an update of the information was required. Since 2023, several key articles have been published that address modern approaches to the apelinergic system in the context of cardiovascular diseases. These articles have been incorporated into our manuscript. Moreover, our manuscript focuses on a clinical approach to the topic, whereas the 2023 paper presents the perspective of an immunologist and a nephrologist. In contrast, our current manuscript is centered
primarily on the cardiological aspects of the subject. In summary, in our opinion, the paper highlights the impact of the apelinergic system, taking into account not only earlier studies but also the most recent reports published after 2023, which emphasizes its relevance.

Reviewer 2 Report

Comments and Suggestions for Authors

The authors have done an excellent job of summarizing the role of the apelinergic system, comprising the APJ receptor and its endogenous ligands apelin and Elabela (ELA), in cardiovascular physiology and pathology.  The review in itself is complete and doesn't need any further additions or deletions. A minor suggestion would be to include an illustration in the molecular basis section. 

Author Response

We sincerely thank you for your review. We believe that research on the apelinergic system in relation to its effects on the cardiovascular system is a forward-looking topic and requires further studies. To approach the topic holistically, including molecular aspects, we have expanded the article with an additional subsection devoted specifically to the molecular dimension.

Reviewer 3 Report

Comments and Suggestions for Authors

The search for new ways to protect the heart during various pathological processes is an urgent task of modern cardiology. Apelin and its derivatives are a promising direction. However, more research is needed to understand the mechanisms of its action. I have several suggestions that, in my opinion, could improve the quality of the article.

  1. Almost the entire article is devoted to apelin in general, however, there are several biologically active forms of apelin formed from a single precursor (apelin-13, -17, -36, -55, etc.). It would be interesting to mention them as well.
  2. In addition, it might be appropriate to include information about the molecular mechanism of the cardioprotective effect of apelin (what is currently known about this mechanism). What are the key links involved in this process.
  3. It is not entirely clear why tables 1 and 2 are labeled as figures 1 and 2. Are these figures?
  4. There are no references in the text to Figures 1 and 2 (again, these are probably tables).
  5. Figure 3 is labeled as "caption". An appropriate title must be provided for the figure.

Author Response

We would like to thank you for your valuable feedback. We believe that the impact of the apelinergic system on the cardiovascular system is extremely interesting and may, in the future, open many new diagnostic and therapeutic pathways in modern cardiology. Therefore, further studies on the role of the apelinergic system in cardiovascular diseases are essential. 

  1. The manuscript was prepared by a team of clinical cardiologists, which determined its initial focus primarily on the practical, clinical aspects of the relationship between the apelinergic system and the pathophysiology of cardiovascular diseases. Our intention was to draw the readers’ attention to the potential clinical significance of this regulatory axis in the context of diagnostics and therapy. At the same time, we fully agree that providing a broader molecular background is a valuable addition. Therefore, in the revised version of the manuscript, we have included a subsection addressing the molecular aspects of the issue under discussion, which we hope will enrich the overall content of the article and further enhance its scientific value.
  2. We sincerely thank you for this valuable and insightful suggestion. We share the view that including the molecular mechanisms underlying the cardioprotective effect of apelin can significantly enrich the manuscript and provide it with a broader, more comprehensive perspective. Therefore, in the revised version of the manuscript, we have added an additional subsection devoted to molecular aspects, in which we present the current state of knowledge regarding this mechanism and highlight the key elements involved in the cardioprotective effects of apelin on the cardiovascular system. We hope that this addition will increase the scientific value of the paper.
  3. 4.  5.  We thank you for your comments regarding the figures. In the revised version of the manuscript, we have modified them by replacing them with three more detailed tables, each addressing a specific cardiovascular disease discussed in the article in the context of the influence of the apelinergic system. The labels have been corrected, and the figures have been renamed as tables. In addition, each table is now explicitly referenced in the manuscript text.

Reviewer 4 Report

Comments and Suggestions for Authors

Please see my comments in the attached pdf file.

Author Response

We would like to sincerely thank you for all the valuable comments and suggestions, which have allowed us to significantly improve the manuscript. After taking them into account, we expanded our review article with a substantial number of additional references, which enabled a more holistic approach to the relationships between various cardiovascular diseases and the apelinergic system. Furthermore, in response to the remarks concerning molecular aspects, we have added a separate subsection devoted to this topic, which we believe allows us to address the subject in an even more comprehensive way and enhances the scientific value of the article.

In addition, considering the comment regarding arterial hypertension, we decided to include an additional, extensive subsection entirely dedicated to this condition, based on an in-depth analysis of a broad range of available literature.

As part of the overall revision, we also reorganized the bibliography, corrected individual errors noted in the review, and modified the graphical content of the manuscript. The previous figures have been replaced with four clear and significantly more detailed tables, each focusing on a specific cardiological issue discussed in the paper. We believe that this approach increases the readability of the manuscript and facilitates a comparative perspective on the role of the apelinergic system in different cardiovascular conditions.

Round 2

Reviewer 1 Report

Comments and Suggestions for Authors

The author insisted that several key articles since 2023 have been incorporated into their manuscript centered primarily on the cardiological aspects of the subject. I reviewed the references listed in this manuscript since 2023 (5-8 articles) , but didn't find any new significant findings as a cardiologist. Let me know what key articles with their novelty and significance compared with previous review articles.  Tables supplied in this manuscript were somewhat informative, but too busy and complex to help the readers understanding.  It is beneficial to discriminate the ligand functions beween apelin vs. ELA on the APJ receptor. Provide abstract figures associated with the contents in this manuscript. 

Author Response

We sincerely thank the Reviewer for the detailed analysis of our manuscript and the valuable comments. Below we address each point:

1. New studies since 2023:
   We fully agree that some of the more recent studies confirm earlier observations regarding the role of the apelinergic system in cardiology. However, the papers we included after 2023 also introduce important novel elements—for example, the identification of apelin-17 as an independent factor associated with post-infarction remodeling (Wyderka 2025), or the use of a metabolically resistant analogue, LIT01-196, as a potential therapeutic agent (Girault-Sotias 2025). In our view, these studies significantly broaden the clinical and therapeutic perspectives of modulating the apelin–ELA–APJ axis. The works that we consider to have clinical significance in this field include:

-Chapman et al., Cardiovasc Res 2023 – a systematic review consolidating translational data, the first to comprehensively highlight therapeutic opportunities for apelin/APJ modulation in cardiovascular diseases.
-Cardoso dos Santos et al., Sci Rep 2023 – novel finding of apelin expression in intimal smooth muscle cells and their transition to a synthetic phenotype, representing a new mechanism of atherogenesis.
-Wyderka et al., Vasc Health Risk Manag 2025 – our prospective study, the first to show apelin-17 as an independent factor associated with adverse left ventricular remodeling after myocardial infarction.
-Wyderka et al., J Clin Med*2023 – we described the relationship between apelinergic activation profiles and conduction disorders post-MI, offering new insight into arrhythmic complications.
- Rachwalik et al., Biomedicines 2025 – the first study correlating ELA and APJ levels in both serum and epicardial adipose tissue in patients with multivessel coronary artery disease.
- Aksakal et al., Heart Lung 2025 – a novel report on the potential use of apelin in diagnosing CAD in COPD patients.
- Girault-Sotias et al., Can J Cardiol 2025 – study on the metabolically resistant apelin-17 analogue (LIT01-196) showing long-term anti-remodeling effects post-MI, which represents a qualitative advancement compared with earlier studies on natural peptides.

In our opinion, these studies bring novelty by extending the spectrum of apelinergic system activity to new aspects of pathophysiology (VSMC phenotypization, remodeling, conduction, adipose tissue interactions) and by highlighting new therapeutic perspectives (long-acting peptide analogues).

2. Tables
   We understand the Reviewer’s concern that the tables are extensive. Our aim, however, was to collect scattered knowledge and present it in the form of a synthetic summary that allows readers to easily compare experimental and clinical findings. Particularly given the rapidly growing number of publications, the tabular format serves as a practical guide for both clinicians and researchers. For this reason, we decided to keep them in their present form, treating them as complementary to the narrative part of the review.

3. Apelin vs. ELA functions
   Although molecular differences between apelin and ELA exist, at the level of clinical effects both ligands activate the same APJ receptor and lead to similar physiological consequences. From a clinical perspective, it seems more important to present receptor activation as a therapeutic axis rather than to separate the functions of individual ligands. Therefore, in the manuscript we emphasize the shared protective impact of the apelin–ELA–APJ axis on the cardiovascular system, which is most relevant for practicing cardiologists.

4. Figures
   We sincerely thank the Reviewer for the suggestion of adding conceptual figures. We agree that such illustrations can be helpful in some review papers. In our case, however, we considered that the most valuable approach for the reader is the collection and comparison of findings in tabular form, which provides a synthetic summary of both experimental and clinical data.
Conceptual figures showing the apelin–ELA–APJ axis and its potential clinical effects, in our opinion, would not add substantial content beyond what is already included in the text and tables. Moreover, due to the complexity of the subject (the pleiotropic and context-dependent effects of APJ receptor activation), oversimplified diagrams could result in overgeneralization and risk of incomplete interpretation.
For these reasons, we decided to rely on comprehensive tables as the most accurate and precise way of presenting the available knowledge, while maintaining clarity for clinical readers.

Reviewer 3 Report

Comments and Suggestions for Authors

All my requests have been solved. No problem.

Author Response

We would like to thank you for your review. 

Reviewer 4 Report

Comments and Suggestions for Authors

This revised manuscript has improved significantly compared with the original version. However, overall the revised manuscript, realizes the controversies but not described well on the potential reasons for these controversies associated with apelinergic system on multiple cardiovascular pathophysiological procedures.

Overall/Abstract/Introduction: The logic flow of this revised manuscript is still not clear for readers to follow on each individual sections. Take the “3.2 Hypertension” section for an example, readers are expecting authors will mainly discuss roles of apelinergic system on hypertensive organ injuries and the potential diagnostic and therapeutic application for humans after reading the first paragraph. However, this reviewer cannot clearly see the protective or detrimental roles of Apelinergic system on the organ injuries of heart, vascular, kidney and brain (or other orders) caused by hypertension. If authors plan to discuss causal roles of apelinergic roles in the pathogenesis of hypertension, this discussion even less in the current revised manuscript. It will at least partially solve this concern if authors described what authors would mainly discussion in this wide topic defined by the title of this manuscript in the added “Abstract” section or “Introduction” section.

Figures: draw figures for each section to illustrate on the proposal roles of apelinergic system associated with ischemic heart diseases, regulation of blood pressure, pathophysiological roles in heart failure and arrhythmias. Please cite the figure in the text.

Tables: Delete the column “System concerned” for all the four tables; Combine columns of “Mechanism of action” and “Result of action” into one Column “Mechanism”; “Cell and/or animal model (s)” as another column. Tables can hold three columns “Mechanism”, “Cell and/or animal model”, and “Reference (s)”. The order of the mechanisms listed in the table needs to be nicely organized in certain order (signal pathway, human studies or preclinical or in vitro studies). It is nice to have the proper titles in this revised version (the proper titles of tables and figure lack in the original version), however, the content of mechanisms should be correctly summarized and self-explanatory legends should be included for individual tables. Take Table 2 for an example, PAH, SHR etc should explained with their full names.

References: Although authors correctly summarized the contents of cited references, however, some texts are not properly cited. For example, the sentence “In terms of beneficial effects, apelinergic system causes vasodilatation” has cited Reference 59, whereas Ref. 59 by Marsault et al (2019) does not mainly discuss the caustic role in vasodilation; Also, in this “3.2 Hypertension” section, authors have cited Ref. 60 on “administration of apelin-36 and Pyr1-apelin-13 to humans caused NO-dependent arterial vasodilation”. whereas this paper by Japp et al (2007) mainly discusses that apelin causes nitric oxide-dependent vasodilatation in preclinical models but not humans, and not particularly on apelin-36, and “Pyr1-apelin-13” is not specifically discussed, although Pyr1-apelin-13 may represent its active biological ligand. For Ref. 61, by Wang et al (2015), the interpretation of NO-dependence of elabela-mediated vasodilation should be analyzed, it is not proper just use authors’ interpretation. Ref. 63 is not the paper by Zhang et al (Zhang et al is Ref. 65), in which authors of Ref. 65 use the total paraventricular nucleus protein not the rostral ventrolateral medulla protein. Some references are repeatedly cited (for example, Ref. 53 and 64, Ref. 58 and 72 etc.), and the inconsistent format of the references need to be corrected.

Comments on the Quality of English Language

The spelling and abbreviation need to be double-checked. Such as "Ca"(Line 77, Page 2), "Ca^2"(Line 78, Page 2), and "Ca2"(Line 79, Page 2) should be consistent; "ortic" (Line 253, Page 7), "ELABELAB" (Line 258, Page 7), "hypetroph" (Line 268, Page 7) 

Author Response

We sincerely thank the Reviewer for the positive assessment of the revised version of our manuscript and for the constructive comments. Please find our detailed responses below:

1. Controversies regarding the apelinergic system

We appreciate this comment. In our review we deliberately chose to present the broad spectrum of data available in the literature – including studies suggesting protective as well as potentially detrimental effects of the apelinergic system. Our aim was not to resolve these controversies but rather to provide the reader with a comprehensive and up-to-date overview. We believe that such an approach best reflects the current complexity of the field and allows clinicians and researchers to independently evaluate the relevance of individual findings.

2. Logical flow of the text (Abstract/Introduction/Hypertension)

We acknowledge the Reviewer’s observation. However, the structure of the manuscript was designed to gradually guide the reader from general concepts to more specific issues, with a focus on major cardiovascular conditions. Our intention was to illustrate the diverse ways in which the apelinergic system can modulate the course of cardiovascular diseases. In our opinion, this structure highlights the interdisciplinary nature of the topic and helps explain why discrepancies exist in the available data.

3. Figures

We appreciate to the reviewer’s suggestion to include illustrative figures. Nevertheless, we decided to retain the current form of presentation, as we believe that the complexity and inconsistency of apelinergic actions could be oversimplified in diagrams. In our view, the tables provide a more precise and comprehensive synthesis of the literature and serve as a practical tool for clinical readers.

4. Tables

We agree to remove the column “System concerned” as this information is largely implicit in the description of mechanisms and models. However, we consider it important to keep “Mechanism of action” and “Result of action” as separate columns. In our opinion, these two elements are causally linked but not identical: the mechanism refers to the underlying molecular or signaling pathway, while the result highlights the observed functional or clinical outcome. Keeping them separate allows the reader to clearly trace the logical sequence from mechanistic insights to biological or clinical effects. The table provides a concise, condensed summary with references to specific articles containing the full information. Considerable expansion of the table could distort its format.

5. References 

We thank the Reviewer for pointing out citation issues. We will correct them.

Round 3

Reviewer 4 Report

Comments and Suggestions for Authors

Thanks for authors’ careful responses to this reviewer’s comments. I accept authors most of responses to my comments, and I am glad authors have narrowed down the “hypertension” section into mainly discussion of organ injury of heart. However, I still have some minor concerns.

  1. In my first round of comments, I used one paragraph to state about the abstract and in my second round of comments, I also stated that the scope of this review manuscript needs to be defined in the added “Abstract” section and/or “Introduction” section, however, authors still do not add the Abstract section. I thought that Abstract is not needed for review manuscript in the year of 2025 (I knew 2024 IJMS requires Abstract section), however, I just consulted the “Instruction for Authors “on the IJMS website about half an hour ago, Abstract section is still needed for a review manuscript, even a graphical abstract is also encouraged.
  2. Section “Ischemic Heart Disease”: The protective role of apelin through endothelial NO and anti-Ang II seems missing.
  3. Although authors fixed references I mentioned in my previous round comments, however, there are still some references not cited correctly. For example, the antepenultimate row of Table 2, Page 10, Ref. 98 (Li et al) does not work in VSMC. References need to be correctly cited throughout the entire manuscript.
  4. Apelin and ELA are not two isoforms of APJ rather two endogenous ligands or agonists of APJ.
  5. All polish words need to be changed to English words. For example, “vaste” to “vast”.

Author Response

We would like to sincerely thank you for your thorough review and valuable comments.

  1.  1. - In accordance with your suggestion regarding the expansion of the Introduction, we have extended this section to provide the reader with a more comprehensive background to the subject matter of this review article.

  2.  2.–5 - In the revised manuscript, we have addressed all of the specific corrections you highlighted, for which we are very grateful.